# Caveolin-1 and Atherosclerosis: Regulation of LDLs Fate in Endothelial Cells

**DOI:** 10.3390/ijms24108869

**Published:** 2023-05-17

**Authors:** Alessandra Puddu, Fabrizio Montecucco, Davide Maggi

**Affiliations:** 1Department of Internal Medicine, University of Genoa, Viale Benedetto XV, 6, 16132 Genoa, Italy; fabrizio.montecucco@unige.it (F.M.); davide.maggi@unige.it (D.M.); 2IRCCS Ospedale Policlinico San Martino Genoa, Italian Cardiovascular Network, Largo Rosanna Benzi 10, 16132 Genoa, Italy

**Keywords:** caveolae, caveolin-1, atherosclerosis, LDL

## Abstract

Caveolae are 50–100 nm cell surface plasma membrane invaginations observed in terminally differentiated cells. They are characterized by the presence of the protein marker caveolin-1. Caveolae and caveolin-1 are involved in regulating several signal transduction pathways and processes. It is well recognized that they have a central role as regulators of atherosclerosis. Caveolin-1 and caveolae are present in most of the cells involved in the development of atherosclerosis, including endothelial cells, macrophages, and smooth muscle cells, with evidence of either pro- or anti-atherogenic functions depending on the cell type examined. Here, we focused on the role of caveolin-1 in the regulation of the LDLs’ fate in endothelial cells.

## 1. Introduction

Caveolae are plasma membrane microdomains which appear as flask-shape invagination of about 50–100 nm [1,2] mainly composed of cholesterol, sphingolipids, and caveolins. Caveolins are a family of integral membrane proteins composed of three members with similar structure, but different tissue localization: caveolin-1 (Cav-1) and caveolin-2 are expressed ubiquitously, whereas caveolin-3 is considered the muscle-specific isoform of caveolin [3,4]. In particular, caveolin oligomerization seems to be the driving force for the assembly of caveolae. As a consequence, other caveolar components are recruited to form a dynamic platform that regulates several cell pathways, including lipid homeostasis, endocytosis, intracellular transport, and signal transduction [3,4].

Cav-1, which is the main protein component of caveolae in several tissues, is an integral membrane protein of about 22–24 kDa [4]. Cav-1 is a hairpin-like structure composed of a transmembrane hairpin loop, a juxtamembrane domain which is responsible for the oligomerization, and a regulatory amino-terminal tail that contains a phosphorylatable site [5]. In particular, the juxtamembrane domain acts as a scaffolding protein, allowing the interaction with several signaling molecules. As a consequence, Cav-1 may regulate several cell pathways through the binding between the Cav-1-scaffolding domain and a conserved sequence enriched with aromatic residues [ΦXΦ XXXXΦ, ΦXXXXΦXXΦ, and ΦXΦXXXXΦXXΦ (Φ = aromatic residue, X = any amino acid)] which is present in several signaling and structural proteins [5]. Moreover, the phosphorylation of Cav-1 at pY14-Cav-1 regulates the spatial organization of Cav-1 molecules within the oligomer, allowing the formation of caveolae and their internalization [6].

Caveolae and Cav-1 are involved in several cardiovascular diseases, including atherosclerosis [7,8]. Atherogenesis has been widely described as a low-grade inflammatory disease, at least in its chronic development [9]. One of the earliest events occurring in atherosclerosis is the accumulation of Low-Density Lipoproteins (LDLs) in the intima [10]. Here, LDLs may undergo oxidative modifications, becoming strongly pro-atherogenic and starting the morphological changes that lead to the formation of the fatty streak [10]. Traditionally, after LDL infiltration within the sub-intimal space and endothelial activation, monocyte/macrophages, T lymphocytes, as well as neutrophils, and B cells can infiltrate the atheroma contributing to their formation. Then, smooth muscle cells as well as anti-inflammatory phenotypes of macrophages (i.e., M2) and Treg counteract intraplaque inflammation, releasing anti-inflammatory mediators potentially stabilizing the atherosclerotic lesion [9]. From their initiation, atherosclerotic plaques can evolve in response to still partially clarified mechanisms to stable or less stable phenotypes [9]. Additionally, inflammatory cells and mediators might infiltrate or proliferate within plaques under different microenvironments, potentially determining a different vulnerability in the plaque maturation phases [11]. Some inflammatory biomarkers have been investigated to identify plaques at higher risk of rupture/erosion and consequent acute ischemic complications [12]. Interestingly, it has also been proposed that levels of Cav-1 expression may be related to plaque progression [13,14,15]. However, no useful candidates have been validated so far. Strong limitations still exist, mainly due to low sensibility and specificity of both intraplaque and circulating mediators. On the other hand, atheroma has been shown to be very heterogeneous tissues [16]. Within different plaque portions (for instance upstream versus downstream the blood flow), inflammatory cells and soluble mediators can be differently expressed, suggesting that, within a single plaque, certain portions might be more vulnerable [16]. This intriguing hypothesis, already demonstrated by human studies, may highlight the very high complexity of the “vulnerable plaque” that was already included in the paradigm the “vulnerable blood” and the “vulnerable myocardium” [17]. To clarify this concept, in this review, we will focus on recent evidence for caveolae and Cav-1 as potential atherosclerotic factors and future potential clinical tools.

### 1.1. Caveolae, Caveolin-1 and Atherosclerosis

Caveolae and Cav-1 are involved in several steps that lead to the formation of atheroma [18], probably because Cav-1 is expressed in cells involved in the development of atherosclerosis, including endothelial cells, macrophages, and smooth muscle cells.

Studies on the role of Cav-1 in atherosclerosis have led to conflicting results depending on the cell type considered, demonstrating either a pro- or anti-atherogenic role. However, previous studies concord that the genetic depletion of Cav-1 protects against atherosclerosis [19], supporting a predominant pro-atherogenic role of Cav-1 in the progression of atherosclerosis [20]. On the other hand, it has been observed that the expression levels of Cav-1 in atherosclerotic plaques are inversely correlated with disease severity [13,14,21], and that the decrease in Cav-1 expression is correlated with plaque vulnerability [14], suggesting that Cav-1 levels may indicate plaque progression.

Endothelial cells are one of the main actors in atherosclerosis [22]. They form a barrier between a blood vessel and the surrounding tissue and are responsible of the uptake and transcytosis of LDLs. Therefore, endothelial cell dysfunction is considered one of the main events in atherosclerosis progression.

Caveolae and Cav-1 regulate several functions of endothelial cells, including nitric oxide production, autophagy, cholesterol homeostasis and mechano-transduction (Table 1) [3].

Due to the abundance of Cav-1 in endothelial cells and considering that specific re-expression of Cav-1 in the endothelium recovers atherosclerosis in Cav-1 deficient mice, several studies have been focused on the association between Cav-1 and endothelial cell dysfunction. Although endothelial cell dysfunction is often strictly referred to as decreased production or bioavailability of nitric oxide, several other events result in deleterious effects on endothelial cell function, ranging from the alteration of redox balance to chronic inflammatory response [22].

The involvement of Cav-1 in the regulation of NO production has been deeply investigated. Firstly, it has been observed that the disruption of caveolae with cholesterol depleting agents leads to the decreased production of NO and vessel relaxation [23]. Indeed, the localization of eNOS in caveolae seems to be required to produce NO [24]. Then, it has been demonstrated that eNOS is closely associated with caveolae by interaction with the scaffolding domain of Cav-1 [24]. This interaction results in the inactivation of eNOS, through the blockage of the eNOS calmodulin binding site [25]. After stimulation, the enzyme is activated by the increased Ca^2+^ concentration and the recruitment of regulatory proteins, which lead to the dissociation of the heteromeric complex between eNOS and Cav-1, but not from the caveolae complex [25,26]. Due to this regulatory mechanism, protection from atherosclerosis in Cav-1 deficient mice has been attributed to the loss of the inhibitory effect of Cav-1 on eNOS activity. However, recent studies demonstrated that this atheroprotection was independent of increased NO production [27]. Indeed, plaque formation is reduced even in absence of eNOS in mice lacking the expression of Cav-1 and LDLR [27].

Endothelial dysfunction has also been associated with defective autophagy in endothelial cells. Under physiological conditions, autophagy contributes to the maintenance of endothelial function by decreasing oxidative stress, inducing eNos expression and NO production, and inhibiting the expression of inflammatory cytokines [28,29,30]. Conversely, defective autophagy may promote inflammation and apoptosis, thus contributing to the development of atherosclerotic plaques. It has been shown that the absence of Cav-1 attenuates the initiation of atherosclerosis by promoting autophagic flux [31]. Several studies reported that Cav-1 may influence autophagy in endothelial cells by affecting the expression and intracellular localization of autophagic proteins. The depletion of Cav-1 increases the expression of the autophagic marker LC3BII [31]. On the contrary, increased expression of Cav-1, induced by treatment with high glucose, suppressed the expression of LC3B-II. Interestingly, phagocytosis is suppressed through the interaction of Cav-1 with LC3B and autophagy related protein 5 (ATG5), a protein involved in the autophagy machinery, in the lipid raft [32]. In particular, interaction between Cav-1 and LC3B may lead to opposite effects: the association through the scaffolding domain of Cav-1 in the plasma membrane inhibits autophagy, whereas the association through the intramembrane domain of Cav-1 promotes it [31]. Moreover, Cav-1 suppresses autophagy by sequestering ATG5 in caveolae, thus avoiding the formation of the autophagosome. In addition, the cellular distribution of Cav-1 may be affected by autophagy. Indeed, during the activation of autophagy, Cav-1 translocates from the plasma membrane to the intracellular membrane [32]. Since Cav-1 is degraded in autophagy, the interaction between Cav-1 and LC3B in the cytoplasm through the intramembrane domain contributes to mediating the autophagic degradation of Cav-1. Therefore, this interaction also regulates the amount of Cav-1 used to form caveolae.

The effects of Cav-1 levels on atherosclerosis may also be related to its involvement in cholesterol control. Indeed, Cav-1 may affect levels of cholesterol at various levels. Firstly, Cav-1 may affect cholesterol homeostasis by mediating LDLs endocytosis: once internalized by endothelial cells, LDL particles can be used to ensure cellular cholesterol requirements by releasing free cholesterol through the lysosomal degradation of LDLs [33]. When intracellular cholesterol levels increase, Cav-1 mediates the transport of cholesterol from the endoplasmic reticulum to the cell surface [34,35]. Then, cholesterol has been transferred from caveolae to HDL/Apo-A1, which transfers cholesterol to the liver [36]. Indeed, caveolae and Cav-1 are involved in cholesterol and cholesterol ester exchange between the endothelial cell surface and lipoproteins [34]. In turn, cholesterol levels may regulate the expression of Cav-1, since the depletion of LDL-derived cholesterol induces Cav-1 transcription in endothelial cells [37]. On the other hand, cholesterol synthesis was reduced in Cav-1 depleted cells [34], whereas the increment in Cav-1 expression has been positively associated with the increased uptake of cholesterol ester from HDL [34]. Therefore, intracellular levels of cholesterol and Cav-1 expression are mutually regulated in order to maintain cholesterol homeostasis. Babitt et al. demonstrated that Cav-1 co-localizes with SR-BI, [38], an 82-kDa monomeric protein that mediates the transfer of lipid from lipoproteins into cells via selective cholesterol [39]. SR-BI is known to be the first receptor for HDL to be described [39] and is also considered a scavenger receptor able to recognize both native and modified LDLs [40]. However, the influence of Cav-1 on SR-BI activity is controversial: it has been reported that the co-expression of SR-BI and Cav-1 increased selective cholesterol uptake in macrophages [41], whereas it seems that Cav-1 expression did not affect SR-BI-mediated selective cholesterol uptake and cholesterol efflux in other cell types [34,42]. Furthermore, the expression of Cav-1 in the hepatic cell line HepG2 induces the dimerization of SR-BI and results in decreased selective cholesterol uptake from LDLs favoring those from HDL [43]. Interestingly, the expression of SR-BI has been negatively correlated with atherosclerosis in mouse models, and the anti-atherogenic effects of SR-BI overexpression are supposed to be related to the lowering levels of cholesterol in LDLs [44].

Finally, it has been reported that several endothelial cells in atherosclerotic plaques are characterized by senescence [45]. Interestingly, the expression of Cav-1 is upregulated during aging and is involved in stress-induced premature senescence [46]. The rise of Cav-1 in endothelial senescent cells may increase oxidative stress [46], which is one of the main contributors of endothelial cell dysfunction. Moreover, since Cav-1 mediates increased vascular permeability induced by oxidative stress [47,48], it may contribute to a further increase in endothelial cell dysfunction.

It is well known that, once internalized by endothelial cells, LDL particles can follow two pathways: endocytosis to respond to cellular cholesterol requirements or transcytosis across endothelial cells [33]. Beyond internalization, Cav-1 also contributes to the transcytosis of LDLs in the sub-endothelial space, where they accumulate starting the formation of atheroma. Recent evidence also supports an important role of Cav-1 in determining the degradation of the LDL receptor (LDLR) [49,50], thus affecting the ability of the cells to uptake LDLs. Therefore, the main mechanism responsible for the pro-atherogenic role of Cav-1 may be relative to the regulation of LDLs fate.

### 1.2. Cav-1 and LDL Uptake

LDLs uptake is mediated by several receptors which have been found to co-localize in caveolae [34,38,51,52].

The receptor-mediated endocytosis of LDL was first described by Brown and Goldstein [53]. The classical explanation of these pathways contemplated that LDLR mediates the endocytosis of LDLs and their lysosomal degradation, with the consequent production of free cholesterol and fatty acids; after that, LDLRs are recycled to the cell surface [53]. LDLR is a 160-kDa protein with an extracellular ligand domain formed by seven cysteine-rich repeats of approximately 40 amino acids responsible for the uptake of LDLs [54]. The expression of LDLR is downregulated due to the availability of new cholesterol [55]. Indeed, LDLR expression is regulated by a feedback mechanism that involved the sterol-responsive element (SRE)1-binding proteins (SREBP): cholesterol depletion activates SREBP, which translocates to the nucleus and promotes the transcription of genes encoding LDLR and involved in cholesterol biosynthesis [56]. Moreover, SREBP-1 can also bind the promoter region of Cav-1, leading to cell type-specific activation of Cav-1 gene transcription [57]. These findings suggest that there is a positive correlation between the expression of LDLR and Cav-1.

One of the first studies regarding LDLR saturation reported that the increased concentration of LDL did not result in enhanced endocytosis [33]. In order to quantify the activity of the LDL receptor, in 1986 Meddings et al. provided an equation that defines the relationship between the production of LDLs, numbers of LDLRs, and LDLs concentration in plasma [58]. Although it was difficult to determine parameters used in their equation, they conclude that the rate of degradation of LDLs is correlated with levels of LDLs in plasma, and that the loss of LDLR activity corresponds to the increased production of LDLs [58]. Interestingly, several years later, Pavlides demonstrated that the kinetics of LDLs transport depends on the presence or absence of Cav-1 [59]. This study provided the first evidence of a direct role for Cav-1 in the regulation of LDLs uptake; indeed, silencing Cav-1 expression was sufficient to inhibit the endocytosis of LDLs in endothelial cells. These results may also account for previous evidence that the loss of Cav-1 reduced the LDLs’ infiltration into the artery wall [21]. Recently, it has been reported that Cav-1 may control the LDLs’ uptake by regulating levels of LDLRs on plasma membrane [50]. Once internalized with LDL, LDLR may be recycled into the plasma membrane or degraded by lysosomes. The latter way is mediated by a mechanism dependent on the proprotein convertase sublisin Kexin 9 (PCSK9), a serine protease mainly produced by the liver [49]. Once released in plasma, PCSK9 has a short half-life and it may circulate freely, and so be able to bind to the EGF-A domain of LDLRs, or as a part of LDLs, thus having a lower ability to associate with LDLRs [49,60]. Briefly, when LDLR is not bound to PCSK9 it is probably internalized through a clathrin pathway, the conformational change of LDLR within the resulting endosome results in the dissociation from LDL and its recycling into plasma membrane [49]. On the other hand, when LDLR is bound to PCSK9, it is internalized through a caveolin dependent pathway [50]. In this case, binding to PCSK9 avoids the conformational change of PCSK9 and targets it for lysosomal degradation, leading to lower cell surface level of LDLRs, thereby affecting LDL metabolism. Jang et al. demonstrated that Cav-1 is necessary for the PCSK9-induced degradation of LDLRs [50]. It has been shown that PCSK9 needs to interact with the cyclase-associated protein-1 (CAP-1) to cause the degradation of LDLR [50]. Indeed, the downregulation of CAP-1 expression decreased both the endocytosis of LDLR/PCSK9 complex mediated by Cav-1 and the PCSK9-mediated degradation of LDLRs. Interestingly, CAP-1 directly binds to Cav-1, and LDLRs were not targeted to degradation by PCSK9 in caveolin-deficient cells [50]. Therefore, the involvement of Cav-1 in the PCSK9-mediated degradation of LDLRs may account for previous evidence that LDLs degradation was more efficient in hepatic cells overexpressing Cav-1 [43].

**Table 1 ijms-24-08869-t001:** Summary of roles of Cav-1 in endothelial cell function.

Endothelial Cell Function	Role of Caveolin-1	References
eNOS Activity	Interaction between eNOS and Cav-1 results in inactivation of eNOS	[23,24,25,26]
Autophagy	Cav-1 affects expression of LC3BII	[31,32]
	Cav-1 affects intracellular localization of ATG5	
	Absence of Cav-1 promotes autophagy	
Cholesterol Control	Cav-1 is involved in cholesterol exchange between lipoproteins and plasma membrane	[34,35]
	Cav-1 may regulate intracellular cholesterol levels by mediating LDL endocytosis	
	Cav-1 mediates transport of cholesterol from ER to plasma membrane	
Senescence	Cav-1 expression is increased in senescent cells. Cav-1 is involved in degradation of LDLR	[45,46]
LDLs Trafficking	Rise in Cav-1 expression increases oxidative stress and vascular permeability	[46,47,48,49,50,59]
	Cav-1 is involved in endocytosis and transcytosis of LDLs	
	Cav-1 is involved in degradation of LDLR	

Cav-1 is also involved in the endocytosis of LDLs mediated by activin-like kinase receptor 1 (ALK-1) [52], a TGF-β type I receptor that binds to apoB-100-containing lipoproteins [61]. In 2008, Santibanez et al. demonstrated that ALK1 is located in caveolae in endothelial cells [51]. The internalization of LDLs mediated by ALK-1 occurs independently of its kinase activity [61]. However, Tao et al. demonstrated that endocytosis the of LDLs mediated by ALK-1 is reduced when Cav-1 expression is downregulated [52], suggesting that Cav-1 is required for this process. At the same time, a loss of Cav-1 upregulates levels of ALK-1 in the plasma membrane [52]. It has been observed that LDLs induce the time-dependent internalization of both LDLR and ALK-1 [52], leading to reduced levels of ALK-1 in the plasma membrane of endothelial cells. This reduction, in turn, may affect the ability of endothelial cells to internalize LDLs. Considering that Cav-1 is involved in the internalization of LDLs mediated by ALK-1, these findings may contribute to account for atheroprotection due to Cav-1 depletion.

Endocytosis of oxidized LDL (ox-LDL) is one of the hallmark events in the pathogenesis of atherosclerosis. Lectin-like oxidized low-density lipoprotein receptor-1 (LOX-1) is considered as the major receptor of ox-LDLs in endothelial cells [62]. It is a type II membrane glycoprotein that recognized both acetylated LDL and ox-LDLs with its extracellular C-type lectin-like ligand-binding domain [63,64]. LOX-1 acts as a scavenger receptor and has been found to be upregulated in atherosclerosis lesions [62]. Similar to LDL receptors, LOX-1 is localized in caveolae of endothelial cells [65], and its activity is strictly related to membrane cholesterol and caveolae integrity [65,66]. Indeed, employment of methyl-β-cyclodextrin, that disrupts cholesterol-rich membrane domain, as well as exposure to statins and use of caveolae inhibitors leads to loss of LOX-1 function and decreased internalization of ox-LDL [65,66]. Interestingly, treatment with ox-LDLs upregulates expression of both LOX-1 and Cav-1 in a time dependent manner [65,67]. This positive correlation may favor the establishment of a vicious cycle that contributes to the pro-atherogenic role of Cav-1.

Finally, Cav-1 may affect lipoprotein transport and retention in the artery wall, and consequently their uptake from endothelial cells [27]. Indeed, the depletion of Cav-1 increased flow velocity in both “athero-prone” and “athero-resistant” sites of the aorta [27]. Moreover, Cav-1, acting as a mechanotransductor, is also involved in endothelial stiffness, a biomechanical feature related to cellular elasticity. Indeed, Le Master et al. showed that the depletion of Cav-1 abrogated endothelial stiffening induced by the uptake of oxLDLs and by altered flow, as occurs around vessel curvature [68]. Considering that atherosclerotic lesions occur preferentially where the blood flow is disturbed, the absence of Cav-1 may contribute to reducing the accumulation of LDL and inflammatory factors.

### 1.3. Cav-1 and LDL Metabolism

Once internalized, LDLs may undergo degradation or cross endothelial cells to reach the basolateral side. The rate of these events and the modification of LDLs may affect the progression of atherosclerosis. As previously reported, Cav-1 plays an important role in the regulation of cellular cholesterol homeostasis. Therefore, the expression of Cav-1 may represent a key point of control to limit and prevent cholesterol accumulation.

It has been shown that the degradation of LDLs was promoted by endocytosis mediated by LDLR and that autophagy contributes to reducing lipid retention in the vessel wall [69,70,71,72]. However, the internalization of high amounts of LDLs activates a negative feedback loop that downregulates the expression of LDLR, thus reducing the endocytosis of LDLs mediated by LDLR and the amount of LDLs targeted to degradation [71]. At the same time, the increased expression of Cav-1 may enhance the number of LDLs internalized through caveolae. This overload may affect endothelial cell function, contributing to the progression of atherosclerosis. Moreover, Cav-1 may deliver LDLR to lysosome degradation through the interaction with PCSK9, thus further reducing the number of LDLs targeted to degradation [50]. LDL-derived cholesterol may be transported to the plasma membrane to be used as a structural component, or to the endoplasmic reticulum, where it is esterified by acyl-CoA: cholesterol acyltransferase (ACAT) to be stored in cholesteryl ester droplets [73,74]. On the other hand, once in the lysosomes, LDLs can be easily oxidated and, consequently, not efficiently degradated [75,76]. Intracellular accumulation of ox-LDL activates several pathways which contribute to endothelial dysfunction. Firstly, intracellular ox-LDL accumulation promotes the phosphorylation of NF-κB p65, that raises Cav-1 phosphorylation, leading to increased caveolae formation and, in turn, LDLs uptake [77,78]. Moreover, the phosphorylation of Cav-1 may promote translocation into the cytosol, expression and release of the high-mobility group box 1 (HMGB1), thus contributing to the onset of inflammatory response and macrophage recruitment [77,79]. In addition, the activation of NF-kB p65 may lead to the increased secretion of proinflammatory cytokines, including the vascular endothelial growth factor-A (VEGF-A). The release of VEGF-A activates a feedback loop that further enhances NF-kB activation and VEGF-A secretion. Interestingly, we previously demonstrated that Cav-1 depletion reduced basal and IGF-1-induced VEGF-A secretion in retinal pigment epithelial cells [80]. Therefore, it can be hypothesized that the absence of Cav-1 may also reduce VEGF-A secretion in endothelial cells. Moreover, the phosphorylation of Cav-1 contributes to the inactivation of eNOS, favoring the activity of inducible iNOS, and leading to increased oxidative stress [81]. Consequently, the expression of LOX-1 is upregulated and, once again, LDLs uptake is increased, creating a vicious circle that enhances oxidative stress and endothelial dysfunction. Therefore, the protection against atherosclerosis in Cav-1 depleted models may be also related to a reduced oxidative and inflammatory environment.

### 1.4. Cav-1 and LDL Transcytosis

One of the first steps that leads to the development of atheroma is the accumulation of LDLs in the sub-endothelial space. LDLs cross endothelial cell barrier through transcytosis [33,82], and once in the intima are modified, leading to the activation of endothelial cells and monocytes recruitment. Interestingly, some evidence indicated that the transcytosis of macromolecules is mediated by caveolae in endothelial cells [83,84]. Indeed, caveolae can cross the endothelial cells bypassing the lysosomes, thus bringing their intact cargo to the other side of the barrier [83]. Moreover, it has been shown that Cav-1 is critical not only for the endocytosis of LDLs, but also for their transcytosis across endothelial cells [10]. The early studies showed that the absence of Cav-1 prevented the transcytosis of LDL across endothelial cells [20,85], and that the formation of fatty streak lesion was reduced by about 70% in the absence of caveolae [82]. In particular, the amount of Cav-1 seems to be related to the rate of LDL transcytosis; indeed, the upregulation of Cav-1 expression corresponded to the increased transcytosis of LDLs [31,86,87,88]. It has been shown that the treatment of endothelial cells with high glucose concentration enhances LDL transcytosis by increasing the levels of Cav-1 [31,89]. Bai et al. demonstrated that treatment with a high concentration of glucose increased the accumulation of Cav-1 in endothelial cells [31], thus promoting LDL transcytosis and the retention of atherogenic lipid in the subendothelial space. The mechanism through which high glucose increases the stability of Cav-1 is associated with the suppression of its autophagic degradation [89,90]. This causes the accumulation of Cav-1 in the cytosol and increases the number of caveolae favoring LDL transcytosis. In addition, high glucose-induced LDL transcytosis is also inhibited by the overexpression of Sirt6, which increases autophagic degradation of Cav-1 [89]. The increased expression of Cav-1 is also involved in the transcytosis of LDLs induced by angiotensin II, the main effector molecule of the renin-angiotensin system [91]. This occurs by enhancing the expression of molecules involved in the transcytosis of LDLs, including Cav-1, and is prevented by treatment with methyl-β-cyclodextrin [91]. In addition, Zhang et al. found that the increased expression of Cav-1 is involved in the increased rate of LDL transcytosis induced by TNF-α in endothelial cells in vitro [87]. On the other hand, the critical role of Cav-1 in LDL transcytosis is also supported by evidence that the rate of LDL transcytosis mediated by Cav-1 may also be affected by downregulated levels of Cav-1 and by its post-transductional modification. For instance, the depletion of Cav-1 induces autophagy and reverses the elevated LDL transcytosis induced by treatment with high glucose [31]. Conversely, the autophagic degradation of Cav-1 suppressed the transcytosis of LDLs across endothelial cells [31,90]. Moreover, Zhao et al. demonstrated that the acetylation of Cav-1, which targets Cav-1 to autophagic degradation, was able to inhibit the transcytosis of LDL induced by high glucose [89].

Caveolae and Cav-1 are involved in transcytosis of both oxLDLs and LDLs, mainly through the scavenger receptor SR-B1 [87,88], which is considered the main receptor involved in LDLs transcytosis [92]. Indeed, although LDLR mediates transcytosis of LDL though the blood brain barrier, other evidence indicates that transcytosis in systemic circulation is independent of LDLR [33,93,94]. First, it has been observed that mutations that cause loss of function of LDLR are associated with increased plasma levels of LDLs and lead to accelerated atherosclerosis [93,94]. Moreover, the degradation of LDLR had no effect on LDLs transcytosis. In order to identify which other receptors mediate LDL transcytosis, it has been demonstrated that the binding of LDLs to SR-BI and ALK-1 induces their uptake and transcytosis in endothelial cells [61,95,96]. As reported above, both SR-BI and ALK1 localize in caveolae [51,52]. In particular, LDLs bind to certain residues (159–178) of SR-BI and recruit the guanine nucleotide exchange factor dedicator of cytokinesis 4 (DOCK4) leading to internalization and transcytosis [96]. LDLs also directly bind to ALK-1, in a more distinct site than LDLR, and promote transcytosis with a crossing rate that correlates to ALK-1 expression [61].

All this evidence supports that transcytosis of LDLs is mediated by Cav-1 in endothelial cells. It has been observed that the majority of LDLs are transported through endothelial cells, and only a small amount is degraded [97]. The only exception to this general mechanism has been described in brain endothelial cells [97,98]. Indeed, the lipid composition of brain endothelial cells avoids caveolae vesicle formation and trafficking, thus limiting the caveolae-mediated transcytosis [98]. Consequently, the internalization of LDLs results in their degradation, without a chance of transcytosis in brain endothelial cells [97]. Of interest, the expression of Cav-1 is significantly reduced in the brain compared to peripheral endothelial cells [97]. Although these findings show a different management of LDLs by endothelial cells, they confirm the main role of Cav-1 in LDLs transcytosis.

## 2. Conclusions

The biological functions of Cav-1 are related to the regulation of several intracellular pathways and cholesterol metabolism [34]. Here we focused on evidence that Cav-1 plays an important role in mediating the uptake and transcytosis of LDLs across the endothelial cells, thus participating in the initiation and progression of atherosclerosis [34,99] (Figure 1).

The entering of LDLs in the subendothelial space is considered one of the early steps in atherosclerosis. Therefore, strategies that may downregulate this event are of particular importance in preventing atherosclerosis progression. Atheroprotection in Cav-1 deficient models has been attributed to decrease transcytosis of LDLs. It has been reported that LDL receptors are mainly localized in caveolae; therefore, Cav-1 may control their assembling in the plasma membrane, thus affecting LDLs uptake. This is a very important event, because the binding of LDLs to the cell surface receptor is necessary for their internalization, and, eventually, their transcytosis. The downregulation or the absence of Cav-1, the main caveolar protein, may affect the presence and the function of LDL receptors. Moreover, through interaction with the LDLR/PCSK9, Cav-1 may directly control levels of LDLR in the plasma membrane [49]. In this case, the presence of Cav-1 could seem protective because the caveolin pathway targets the LDLR/PCSK9 complex for degradation, thus reducing the ability of endothelial cells to capture LDLs. However, as a consequence, reduced levels of LDLR may increase the concentration of LDLs in the blood, worsening lipid metabolism. In this contest, it has to be considered that the optimal goal should be to capture LDLs, target them with lysosomes, thus avoiding transcytosis of LDLs, and recycle LDLR into the plasma membrane so that it can capture other LDLs to ameliorate lipid profile [49].

Overall, evidence indicate that high levels of Cav-1 in endothelial cells are correlated to atherosclerosis progression. Indeed, the effects of the majority of pro-atherogenic factors result in increased expression of Cav-1. Moreover, the positive correlation between Cav-1 amounts and levels of both LDLR and LOX-1 may be responsible of the prevalence of transcytosis in peripheral endothelial cells. Therefore, strategies to reduce Cav-1 expression may be useful in switching the fate of LDLs to degradation, as occurs in brain endothelial cells [97]. Levels of Cav-1 may be regulated by affecting its transcription, or by post-translational modification. For instance, increased free cholesterol derived from LDLs metabolism inhibits Cav-1 transcription by inducing the binding of SREBP1 to the Cav-1 promoter [57]. Moreover, the increased expression of Cav-1 may be due to the suppression of its autophagic degradation [31]. On the contrary, the acetylation of Cav-1 by Sirtuin-6 targets Cav-1 for autophagic degradation and reduces transcytosis of LDLs induced by high glucose [89]. Therefore, endothelial cells may regulate the expression of Cav-1 through different mechanisms. Considering that autophagy is one of the mechanisms through which endothelial cells regulate the levels of Cav-1, and the reduced inhibitory feedback on Cav-1 expression due to decreased LDLs metabolism, it can be hypothesized that the imbalanced levels of Cav-1 may be one of the first events during endothelial cell dysfunction. Once Cav-1 expression starts to increase, several pathways related to it become more efficient. Consequently, the amplification of signals mediated by Cav-1 leads to the establishment of a vicious circle that contributes to further worsening endothelial cell function, increased number of caveolae, and, in turn, increased uptake and transcytosis of LDLs, and so the rise in oxidative stress and inflammation.

All these considerations confirm the important role of Cav-1 in driving atherosclerosis and suggest that the regulation of Cav-1 expression and of its interactions may be a new goal to be pursued among strategies to counteract LDL transcytosis.

## 3. Future Perspectives

Cav-1 is involved in every step that regulates LDL fate in endothelial cells. In addition, Cav-1 may exert proatherogenic effects by modulating inflammatory cells, both infiltrating and proliferating within plaques. Moreover, the potential different expression of Cav-1 on distinct atheroscleroic districts (i.e., carotid vs. coronary) as well as within different portions of atherosclerotic plaques may account for the different susceptibility of plaque formation. This evidence makes Cav-1 a potential target of new strategies to prevent and downregulate the genesis and progression of atherosclerosis. Therefore, future clinical investigations should be addressed to counteract the contribution of Cav-1 in the progression of atherosclerosis.

## Figures and Tables

**Figure 1 ijms-24-08869-f001:**
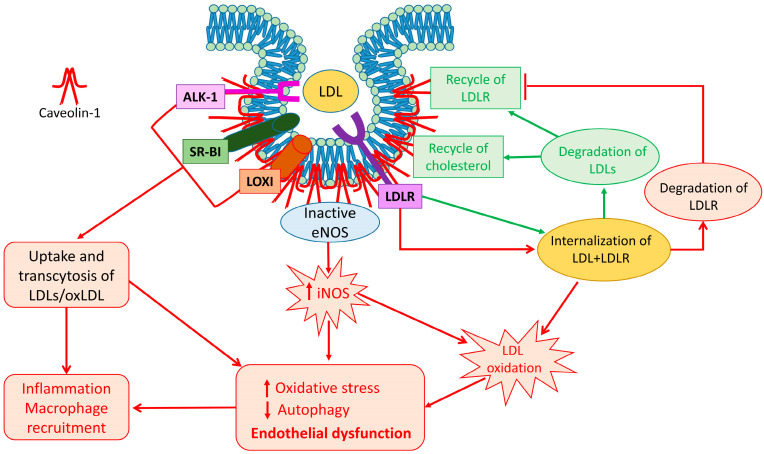
Caveolin-1 and LDL trafficking: a schematic representation of different fates of LDLs after Cav-1-mediated trafficking. Receptors for LDL and oxLDL are localized in caveolae. Their interaction with Cav-1 favors uptake and transcytosis of LDL and oxLDL. When LDL are internalized through LDLR (green lines), they are degraded and cholesterol and LDLR are recycled in the plasma membrane. However, when LDLR interacts with PCSK9, the complex LDL-LDLR is internalized through Cav-1 (red lines), and the LDLR is degraded. Expression of Cav-1 is strictly related to internalization and transcytosis of LDLs. The overload of LDLs, as well the imbalanced activity of eNOS and iNOS, induces LDL oxidation and decreases their degradation leading to increased oxidative stress and attenuation of autophagy, thus contributing to endothelial dysfunction. These events, in turn, activate production and release of inflammatory factors contributing to the establishment of an inflammatory environment and to the recruitment of macrophages, thus favoring the onset and progression of atherosclerosis.

## Data Availability

No new data were created and analyzed in this study. Data sharing is not applicable to this article.

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
