# Peer review of "Caveolin-1 and Atherosclerosis: Regulation of LDLs Fate in Endothelial Cells"

_ijms, 2023, doi:10.3390/ijms24108869_

Round 1
Reviewer 1 Report
In this manuscript, authors focused on the role of caveolin-1 in regulation of LDLs fate in endothelial cells. Their review improved the understanding of the function of caveolin-1 in plaque formation and might benefit the development of corresponding therapeutic strategies. However, authors only described the regulation of caveolin-1 in LDL uptake and transcytosis, while caveolin-1 also played a critical role in autophagy and lipid metabolism in endothelial cells. As a review focusing on the regulation of LDLs fate, it’s recommended to add a few paragraphs about the degradation, metabolism, and related downstream signaling of LDLs regulated by caveolin-1 in endothelial cells.
Author Response
In this manuscript, authors focused on the role of caveolin-1 in regulation of LDLs fate in endothelial cells. Their review improved the understanding of the function of caveolin-1 in plaque formation and might benefit the development of corresponding therapeutic strategies. However, authors only described the regulation of caveolin-1 in LDL uptake and transcytosis, while caveolin-1 also played a critical role in autophagy and lipid metabolism in endothelial cells. As a review focusing on the regulation of LDLs fate, it’s recommended to add a few paragraphs about the degradation, metabolism, and related downstream signaling of LDLs regulated by caveolin-1 in endothelial cells.
Reply: We thank you for the suggestions. We improved the review adding the role of Caveolin-1 in regulation of autophagy in endothelial cells (Section 2.1: lines138-162; and in Section 2.4: lines 364-374 and 383-388). Moreover, we added Section 2.3 regarding Cav-1 and LDL metabolism, in which we explore the role of Cav-1 in LDL metabolism and the intracellular signaling related to LDL uptake.
Reviewer 2 Report
In this manuscript, the authors have summarized the role of caveolae and Cav-1 as potential atherosclerotic factors in the regulation of LDLs fate in endothelial cells.
1. Basically, this review gave some detailed information about the results of former studies but I miss the very latest findings from the last 3-5 years and the authors’ own paper(s) in this topic (https://www.mdpi.com/2075-1729/12/1/44). The role of caveolaes and Cav-1 have been described on similar way in other reviews (https://doi.org/10.1515/mr-2021-0005; DOI: 10.1111/bph.15272; doi:10.1097/MOL.0000000000000701); therefore, the novelty of this review is highly questionable. Adding the very latest results and own findings/hypotheses may improve the quality of the manuscript.
2. The authors mentioned at the end of the introduction that they summarized the potential clinical findings of caveolaes but these results did not describe clearly in the conclusion.
3. Adding the relevant references to the table may improve it.
4. The authors should complete Figure 1 with data about regulation of eNOS activity and cholesterol control, similar in this paper https://doi.org/10.1515/mr-2021-0005. Thus, the reader can see an excellent summarizing and well-organized figure about the topic of the manuscript.
5. English spell and formal check are needed. Abbreviation of proprotein convertase subtilisin/kexin type 9 (PCSK9) is incorrect throughout the text, especially in Ln196-214.
In summary, the lack of novelty and incomplete summarizing figure are the weakness of the manuscript.
Author Response
In this manuscript, the authors have summarized the role of caveolae and Cav-1 as potential atherosclerotic factors in the regulation of LDLs fate in endothelial cells.
- Basically, this review gave some detailed information about the results of former studies but I miss the very latest findings from the last 3-5 years and the authors’ own paper(s) in this topic (https://www.mdpi.com/2075-1729/12/1/44). The role of caveolaes and Cav-1 have been described on similar way in other reviews (https://doi.org/10.1515/mr-2021-0005; DOI: 10.1111/bph.15272; doi:10.1097/MOL.0000000000000701); therefore, the novelty of this review is highly questionable. Adding the very latest results and own findings/hypotheses may improve the quality of the manuscript.
Reply: We thank you for the comment. We revised the manuscript adding the latest findings on the involvement of Caveolin-1 in LDL fate. In particular, we discuss the role of Caveolin-1 in regulation of autophagy in endothelial cells (Section 2.1: lines138-162; and in Section 2.4: lines 364-374 and 383-388); and we added Section 2.3 regarding Cav-1 and LDL metabolism, in which we explore the role of Cav-1 in LDL metabolism and the intracellular signaling related to LDL uptake.
- The authors mentioned at the end of the introduction that they summarized the potential clinical findings of caveolaes but these results did not describe clearly in the conclusion.
Reply: in this review we described the mechanisms through which Caveolin-1 may affect LDL fate in endothelial cells, highlighting its involvement in the onset and progression of atherosclerosis. We summarized this knowledge in the new figure of the conclusion. All the evidence confirms the important role of Cav-1 in driving atherosclerosis and suggests that Cav-1 may be a potential target of new strategies to prevent and downregulate genesis and progression of atherosclerosis. Moreover, we speculated about the hypothesis that the imbalanced levels of Cav-1 may be one of the first event during endothelial cell dysfunction. We hope that these changes contribute to better explain the potential clinical implication of regulation of caveolin-1 expression in preventing up-take and transcytosis of LDLs, and consequently in the establish of a pro-atherogenic environment.
- Adding the relevant references to the table may improve it.
Reply: we thank you for the suggestion, we added references to table 1.
- The authors should complete Figure 1 with data about regulation of eNOS activity and cholesterol control, similar in this paper https://doi.org/10.1515/mr-2021-0005. Thus, the reader can see an excellent summarizing and well-organized figure about the topic of the manuscript.
Reply: As you suggested, we improved the figure adding data about regulation of eNOS activity, cholesterol control and LDL metabolism
- English spell and formal check are needed. Abbreviation of proprotein convertase subtilisin/kexin type 9 (PCSK9) is incorrect throughout the text, especially in Ln196-214.
Reply: we carefully revised and corrected the manuscript.
In summary, the lack of novelty and incomplete summarizing figure are the weakness of the manuscript.
Reply: we hope that the revision improves the manuscript and meets your requests.